# Phototherapeutic Induction of Immunogenic Cell Death and CD8+ T Cell-Granzyme B Mediated Cytolysis in Human Lung Cancer Cells and Organoids

**DOI:** 10.3390/cancers14174119

**Published:** 2022-08-25

**Authors:** Asta Valančiūtė, Layla Mathieson, Richard A. O’Connor, Jamie I. Scott, Marc Vendrell, David A. Dorward, Ahsan R. Akram, Kevin Dhaliwal

**Affiliations:** 1Translational Healthcare Technologies Group, Centre for Inflammation Research, Queen’s Medical Research Institute, University of Edinburgh, 47 Little France Crescent, Edinburgh EH16 4TJ, UK; 2Department of Pathology, Royal Infirmary of Edinburgh, Edinburgh EH16 4SA, UK

**Keywords:** photodynamic therapy, lung cancer, organoids, immunogenic cell death

## Abstract

**Simple Summary:**

Immunogenic cancer cell death and photodynamic therapy play an important emerging role in cancer treatment. Understanding mechanisms involved in tumor killing via immune cells in response to photodynamic therapy is important for developing new anticancer approaches. In this study, we report that methylene blue photodynamic therapy decreases the proliferation of lung cancer cells and patient derived non-small cell lung cancer organoids via immunogenic cells death and granzyme B mediated cytolysis.

**Abstract:**

Augmenting T cell mediated tumor killing via immunogenic cancer cell death (ICD) is the cornerstone of emerging immunotherapeutic approaches. We investigated the potential of methylene blue photodynamic therapy (MB-PDT) to induce ICD in human lung cancer. Non-Small Cell Lung Cancer (NSCLC) cell lines and primary human lung cancer organoids were evaluated in co-culture killing assays with MB-PDT and light emitting diodes (LEDs). ICD was characterised using immunoblotting, immunofluorescence, flow cytometry and confocal microscopy. Phototherapy with MB treatment and low energy LEDs decreased the proliferation of NSCLC cell lines inducing early necrosis associated with reduced expression of the anti-apoptotic protein, Bcl2 and increased expression of ICD markers, calreticulin (CRT), intercellular cell-adhesion molecule-1 (ICAM-1) and major histocompatibility complex I (MHC-I) in NSCLC cells. MB-PDT also potentiated CD8^+^ T cell-mediated cytolysis of lung cancer via granzyme B in lung cancer cells and primary human lung cancer organoids.

## 1. Introduction

Lung cancer is the leading cause of cancer death worldwide and often presents late with poor prognosis and/or poor response to therapies. Treatment options for late-stage disease remain limited, but the advent of immunological approaches have led to significant hope of improving the current dismal prognosis of late stage disease [1]. However, therapeutic approaches face significant hurdles such as the presence of specific DNA mutations, advanced metastasis, drug-resistance, reduced chemosensitivity, lack of activated cytotoxic T cells, and the immunosuppressive cancer microenvironment [2,3].

The survival, proliferation and sensitivity to drug treatment of cancer cells is not entirely cell intrinsic and can be influenced by the cellular composition and environmental conditions in the tumour microenvironment (TME) (Reviewed [4]). The impact of the TME on sensitivity to therapeutic interventions is not accounted for in traditional 2D monocultures of cancer cells where interactions with other cell types and components of the extracellular matrix are absent. Thankfully, advances in 3D culture techniques using cancer spheroids and organoids have provided models more representative of the in vivo situation and are proving their value in drug screening [5] and the development of personalized organoid-guided precision medicine [6]. Non-malignant cells can increase the rates of cancer cell proliferation in organoids and increase their drug resistance compared to the 2D culture monocultures [7,8]. The presence of other cell types in organoid cultures, such as cancer-associated fibroblasts, can also alter the threshold of sensitivity to drug treatment in ways not predicted from 2D culture models [9]. As well as cell to cell interactions and exposure to secreted factors, the physical constraint of 3D culture also promotes epigenetic changes which facilitate migration and invasiveness [10]. As measures of proliferation, invasiveness and viability all form readouts of drug sensitivity, the higher baselines and thresholds offered by 3D-culture systems offer a more realistic platform for testing therapeutic drug treatments. 

Photodynamic therapy (PDT) is an emerging treatment modality for neck, ovarian, breast, NSCLC and malignant melanoma [11,12]. The advent of interventional approaches to access lesions or to treat distant metastases has shown utility in the treatment of different solid cancers, in parallel to surgery, chemotherapy, radiation therapy or immunotherapy [13,14,15,16]. During PDT, a photosensitizer (PS) is delivered followed by illumination with specific light which excites the PS to generate reactive oxygen species, leading to cell death. PDT has been reported to promote apoptosis and necrosis, inducing damage-associated molecular pattern molecules (DAMPs), generating an immune response and increasing antitumor immunity against solid cancer [17,18,19,20]. An advantage of PDT over chemotherapy is that PDT can be used to treat drug-resistant cancer and enhance responses to tumour-derived antigens by promoting ICD [21,22].One of the main mechanisms by which PDT putatively destroys tumours is through the activation of an immune response against cancer cells. The induction of ICD by PDT stimulates antigen presentation by dendritic cells (DCs) and enhances the cytotoxicity of T lymphocytes [23,24]. Building upon recent reports [12,25,26,27], we thus embarked upon characterising the effects induced by MB-PDT (a clinically approved sensitizer) on ICD in NSCLC cells using low energy light sources and investigated the role of activated non-autologous CD8^+^ T cells on survival of MB-PDT-treated human lung cancer cells and patient-derived organoids in co-culture.

## 2. Materials and Methods

### 2.1. Materials and Chemicals

All chemicals, media for growing cells and organoids, antibodies, probes and tools are detailed in Appendix A.

### 2.2. Lung Cancer Cell Lines 

H1299 (P53 null) and A549 (P53 wt) cells, human Non-Small Cell Lung Carcinoma cell lines, were obtained from ATCC (Manassas, VA. USA). The A549 cells have a *Kras* mutation and are less invasive than the HI299 cell line, which lack p53 expression; they differ in their capacity to repair DNA damage due to the ability of p53 to modulate the capacity of HMGB1 to inhibit DNA repair [28]. The cells were grown in complete media (DMEM plus 10% FBS, 100 IU/mL penicillin G sodium salt, 100 µg/mL streptomycin sulphate and 2 mM L-glutamine). Cells were regularly passaged in T25 cell culture flasks upon reaching 80–90% density. The mycoplasma contamination was tested on a regular basis. 

### 2.3. CD8^+^ T Cell Isolation 

Human peripheral blood mononuclear cells (PBMCs) were isolated from healthy donor blood using Lymphoprep^TM^ (catalogue no. 07801, Stem-Cell Technologies Cambridge UK) and following the recommended original company’s protocol. CD8^+^ lymphocytes were isolated from PBMCs using CD8 Micro Beads (catalogue no. 130-045-201, Miltenyi Biotec, Bergisch Gladbach, Germany). CD8^+^ cells were propagated in RPMI 1640 medium containing 10% FBS, 1% penicillin/streptomycin, human IL-2 (210 U/mL), anti-CD3 (1 μg/mL) and anti-CD28 (1 μg/mL) antibodies in 37 °C with 5% CO_2_ incubator for 2–7 days. 

### 2.4. Culture of NSCLC Tumour Organoids

Human lung cancer organoid culture has been described previously [29]. Briefly, NSCLC human tissue was obtained from patients undergoing thoracic surgery for curative treatment of lung cancer. Tissue was digested by mincing the tissue and then incubating in a cocktail of Collagenase IV and DNase for 1 h at 37 °C. Following incubation, the sample was centrifuged at 300 g for 5 min and the supernatant removed. The minced tissue was then incubated with 2 mL TryplE Express for 5 min at 37 °C. DMEM containing 10% FCS was added to stop the reaction. Following centrifugation, the sample was re-suspended in DMEM and vortexed before being passed through a 70 μm cell strainer. The cell suspension could then be counted and seeded for culture. Cell suspension was mixed with Matrigel at a concentration of 1 × 10^6^ cells/mL and 50 μL domes were pipetted onto warm 6 well cell culture plates with 6 domes per well or 30 μL. Domes were pipetted into individual chambers of 8 chamber Ibidi slides (catalogue number IB-80801). Plates/slides were incubated upside down for 10 min at 37 °C to allow domes to set and then 2 mL or 200 μL of organoid complete media was added to each well respectively. 

### 2.5. Photodynamic Treatment for Lung Cancer Cells and Lung Cancer Organoids

Methylene blue (MB) was used as a photosensitizer in this study. The photosensitizer solution was prepared in distilled water at 2 mM (a stock solution) and protected from light. Lung cancer cells were incubated for 2 h with MB (20 μM) in DMEM complete medium at 37 °C with 5% CO_2_ incubator. The cells were irradiated with LED arrays with maximum emission wavelength at 625 nm, corresponding to total light doses of 10 J/cm^2^ in a dark room a dose which we established did not affect viability. Light exposure of up to 60 J/cm^2^ does not decrease viability of A549 cells [30]. Afterwards, the medium containing MB was discarded and fresh complete DMEM medium was added. MB-PDT-treated cells were incubated in humidified atmosphere at 37 °C with 5% CO_2_ for the time required. Control experiments, such as cells neither exposed to the PS nor light (control/untreated), were included in all experiments. 

In case of organoids, the 3D culture was treated with MB (20 μM or 40 μM) and total light doses of 10 J/cm^2^ were used.

### 2.6. Cell Proliferation Assay

Cell proliferation was determined with a WST-1 assay according to the manufacturer’s instructions. H1299 or A549 cells were plated into the 96-well plates the day before the experiment, reaching 70–80% confluence on the day of experiment. The WST-1 reagent was added to the cells and incubated at 37 °C with 5% CO_2_ incubator for 1 h. The absorbance was measured by Synergy H1 Hybrid Reader (BioTek Instruments, Winooski, VT, USA) at 450 nm wavelength. 

### 2.7. Lung Cancer Cells-Lymphocyte Co-Culture Killing Assay and WST-1 Assay 

The co-culture killing assay has been described previously [31]. Briefly, H1299 cells were grown in complete DMEM media at 37 °C. For the killing assay, H1299 cells, as target cells, were seeded into the 96 well plate and treated with MB-PDT (as described above). Activated CD8^+^ T cells, as effector cells, were added into each well with cancer cells at a ratio of 8:1, and cultured for 24 h in DMEM media containing rhIL-2 (320 U/mL), anti-CD3 (4 μg/mL) and anti-CD28 (10 μg/mL) antibodies. Then, the wells were washed twice with PBS, and CD8+ T cells were washed off. Complete DMEM phenol red free media and the WST-1 reagent were added and incubated at 37 °C with 5% CO_2_ for 1 h. The absorbance was measured by Synergy H1 Hybrid Reader (450 nm). In case of using MB-PDT-treated or untreated human lung cancer organoids, we followed the same protocol as described above.

### 2.8. Fluorescence Microplate Reader Experiment

Activated CD8^+^ T cells were added into a culture of H1299 cells (with/without treatment) at a ratio of 2:1 in a completed DMEM-F12 media supplemented with hrIL-2 (320 U/mL), anti-CD3 (4 μg/mL) and anti-CD28 (10 μg/mL) antibodies for one hour in 37 °C incubator using a 96 well plate. Then, Granzyme (GrzB) probe H5 (a final concentration of 5 µM) was added and incubated with the co-culture of living cells until 24 h. In cases where a GrzB inhibitor IV (10 μM) was used, the cancer cells were pre-incubated with the inhibitor for one hour prior to addition of the activated CD8^+^ T cells and the GrzB probe. The fluorescence intensity of GrzB probe H5 was measured by Synergy H1 Hybrid Reader (450 nm Excitation, 510 nm Emission). All experiments were repeated at least three times. 

### 2.9. Cytospin Technique and Microscopy 

CD8^+^ lymphocytes were isolated from the human peripheral blood mononuclear cells (PBMCs) using CD8 Micro Beads (catalogue no. 130-l045-201, Miltenyi Biotec Bergisch Gladbach, Germany) and following the company’s protocol. CD8^+^ cells were propagated from 2 to 7 days in RPMI 1640 medium containing 10% FBS, 1% penicillin/streptomycin, human IL-2 (210 U/mL), anti-CD3 (1 μg/mL) and anti-CD28 (1 μg/mL) antibodies. Then, the CD8^+^ T cell suspension was prepared at 0.5 × 10^6^ cells/mL in 200 µL of PBS 1× and loaded in cuvettes. The cuvettes with cells were spun at 800 rpm for 3 min at room temperature (RT). The slides containing the centrifuged cells were immediately fixed with 4% formaldehyde for 10 min at RT. Following fixation, the cells were blocked and permeabilised with 5% donkey serum and 0.5% triton X-100 in cold PBS for 30 min at RT. The cells were then incubated with the anti-human GrzB (496B) antibody (1:500) for 24 h at 4 °C. After washing with 0.5% tween20 in PBS, the cells were incubated with Alexa Flour 647 anti-Rat (IgG) secondary antibody (1:1000) and Dapi (1:1000) for one hour at RT in the dark. After washing with 0.5% tween20 in PBS for one hour in a room temperature, the slides were prepared for imaging and investigated by SP8 confocal microscope with 20× objective. Fluorescence image reconstruction was conducted by using ImageJ software.

### 2.10. Immunofluorescence for Lung Cancer Cells 

Cells were seeded on glass coverslips and propagated for 24 h. Then, cells were treated and propagated for a further 24 h. Cells were fixed in 4% paraformaldehyde for 15 min, incubated in permeabilisation/blocking buffer (0.2% Triton, 3% goat (or donkey) serum, 3% bovine serum albumin) for 1 h at RT. Calreticulin antibody was diluted (1:200) and incubated overnight at 4 °C before incubation with the secondary antibody (1:1000) for 1 h at RT, and mounted. Cells were imaged on SP8 confocal microscope with 40× objective. ImageJ was used to measure cell fluorescence. The corrected total cell fluorescence (CTCF) was calculated using this formula: CTCF = Integrated Density—(Area of selected cell × Mean fluorescence of background readings). 

### 2.11. Staining Organoids for Confocal Imaging

Organoids were grown in 8 well chamber slides in Matrigel. Media was removed and 100 μl Biolegend fixation buffer was added to each chamber and slides were incubated at 4 °C for 45 min. Fixative was removed and 400 μl cold phosphate buffered tween (PBT: 0.05% (*v*/*v*) Tween-20 in PBS) was added to each chamber and incubated at 4 °C for 20 min. Then, PBT was removed and 200 μl organoid washing buffer (OWB: 0.1% (*v*/*v*) TritonX-100, 0.02% SDS, 0.2% BSA in PBS) added to each chamber to block organoids and incubated for 30 min at 4 °C. PanCK antibody was added (1:100) in 200 μl of OWB to each chamber being stained. This was incubated overnight at 4 °C on a plate rocker. Chambers were washed 3× with OWB then the Goat Anti-Mouse IgG Alexa Fluor 647 secondary antibody added (1:500) in 200 μl solution of OWB. This was incubated for 90 min at 4 °C on a plate rocker. Chambers were washed 2× with OWB. 200 μl OWB, 1 drop ActinGreen ReadyProbe and 1 drop DAPI ReadyProbe was added to the chambers and slide incubated 4 °C for 30 min on plate rocker. Chambers washed 2× with OWB and 200 μl OWB added to chambers before imaging on Leica SP8 with 20× objective.

### 2.12. Flow Cytometry

Cells were washed with PBS and counted. A cell suspension was made of 10^7^/mL in PBS with a flow cytometry buffer (0.2% BSA, 2 mM EDTA). For each primary antibody used, 100 µL of cell suspension (1 × 10^6^ cells) was mixed with specific fluorescently labelled antibodies at their recommended dilutions and incubated for 20 min at 4 °C in FACS tubes protected from light. The cells were washed in flow cytometry buffer twice and centrifuged at 300× *g* for 6 min and fixed with fixation buffer. Just before running the samples on a flow cytometer, the cells were centrifuged at 300× *g* for 6 min and re-suspended in 400 µL of PBS. For all the experiments a minimum of 10,000 gated events were collected; identical conditions were replicated at least three times each. Specifically, for AnnexinV/PI experiments the cells (1 × 10^6^ cells) were prepared according to the manufacturer’s protocol. Samples were analysed on BD Bioscience LSR Fortessa 5 instruments (Franklin Lakes, NJ, USA). Data was acquired, analysed, and plotted using FACS Diva and Flow Jo software. All antibodies are detailed in Appendix A.

### 2.13. Staining Human Lung Cancer Organoids for GrzB probe H5 in a Direct Cell-to-Cell Contact between Organoids and T Cells

Organoids were grown in 30 μl dome of Matrigel per individual chamber of an 8 chamber Ibidi slide. Organoids were incubated for 2 h with MB (20 μM or 40 μM) in complete medium (Appendix A) at 37 °C in a 5% CO_2_ incubator. Then, organoids were irradiated with an LED (625 nm, 10 J/cm^2^) in a dark room. The medium containing MB was discarded, and fresh complete media was added. MB-PDT-treated organoids were incubated in a humidified atmosphere at 37 °C with 5% CO_2_ for 24 h or longer (48 h or 5 days). Then, lung cancer organoids (with/without treatment) were incubated with a recovery solution of 200 μl per well for 45 min at RT. Next, organoids from the well were moved into a tube. After washing twice with cold 1× PBS, and centrifugation at 300× *g* at 4 °C for 10 min, un-dissociated organoids were re-suspended with non-autologous activated CD8+ T cells at a ratio of 1:3, respectively, in complete Advanced DMEM/F12 media containing rhIL-2 (320 U/mL), anti-CD3 (4 μg/mL) and anti-CD28 (10 μg/mL) antibodies, and kept at 37 °C, 5% CO_2_ for two hours. The Granzyme B probe H5 was added to the co-culture at a concentration of 5 μM and incubated for one hour at 37 °C, 5% CO_2_ in darkness. Hoechst 33,342 (nuclei stain) was added to the chambers at the same time. Live imaging was performed on Leica SP8 with 40× objective.

### 2.14. Immunoblotting

Cells were lysed in whole cell lysis buffer for 1 h on ice and centrifugated at 13,000 rpm for 15 min. Protein extractions were quantified by Bradford’s method (Bio-Rad Hercules, CA, USA). Protein samples were separated by SDS-PAGE and transferred to nitrocellulose membrane before blocking with 5% non-fat milk in PBS 1× for 1 h at RT. Primary antibodies (1:1000) were incubated overnight at 4 °C before adding secondary HRP-conjugated antibody (1:2500) for 1 h at RT. The chemiluminescence was used to visualise antigen-antibody complexes. All antibodies are detailed in Appendix A.

### 2.15. Statistical Analysis

Results are expressed as the mean ± SD of the replicates for each group. Statistical analysis was performed using two-tailed Student’s t-test analysis. *p*-value < 0.05 was considered significant. 

### 2.16. Patient Tissue and Blood Donors

The studies involving human lung tissue participants were reviewed and approved by Lothian Regional Ethics Committee (REC) (REC No: 20-HV-069) and NHS Lothian and facilitated by NHS Lothian SAHSC Bioresource (REC No: 20/ES/0061). All healthy blood donors were approved by the Centre for Inflammation Research Blood Resource (CIRBRP004).

## 3. Results

### 3.1. MB-PDT Decreases Lung Cancer Proliferation and Increases Apoptosis

We analysed the effect of MB-PDT on two NSCLC cell lines, H1299 (P53 null) and A549 (P53 wt). In the whole protein lysates, isolated at 6 and 24 h post-treatment, the expression of Bcl2, an anti-apoptotic protein, was significantly reduced in MB-PDT-treated H1299 cells compared to untreated samples (Figure 1A). MB-PDT treatment significantly reduced the proliferation of H1299 (Figure 1B) and A549 cells (Figure 1C) 24 h post-treatment. MB-PDT-treated A549 cells displayed decreased viability compared to untreated cells 24 h post-treatment. Cisplatin (CDDP) and Mitoxantrone Dihydrochloride (MTX) –treated cells showed a decreased viability compared to untreated cells. CDDP was used as a positive control of apoptosis and MTX as a positive control of ICD. In A549 cells treated with MB at 20 µM without PDT, the viability of cells was unchanged (Figure 1D), and the percentage of Annexin V-positive cells was not different compared to untreated cells 24 h post-treatment (Appendix A). The decreased proliferation of MB-PDT-treated A549 cells was associated with late apoptosis (26.4%) compared to untreated cells (0.57%) 24 h post-treatment (Appendix A). In MB-PDT-treated and CDDP-treated A549 cells, activated caspase 3/7 probe activity was increased indicating apoptotic cell death (Figure 1E). 

### 3.2. MB-PDT Induces Cell Death and Disintegration of Human Lung Cancer Organoids

Human lung cancer 3D cultures were established (Figure 2A) and characterised (Figure 2B) to confirm clonal tumour expansion by expression of Pancytokeratin (PanCK), a marker of lung adenocarcinoma. 20 μM MB-PDT caused obvious morphological changes under bright-field microscopy after 48 h (Figure 2C). Confocal imaging also confirmed death of organoids by Toto3-iodide, a marker of dead cells, in MB-PDT-treated organoids 48 h post-treatment (Figure 2C). We did not detect significant cell death of organoids by Toto3-iodide in MB-PDT-treated organoids 24 h post-treatment. Interestingly, the morphology of 40 μM MB-PDT-treated organoids showed dramatic changes (loss of 3D structure), and demonstrated disintegration after sustained treatment (5 days). 40 μM MB-PDT induced death in lung cancer organoids in a dose-dependent manner detected by Toto3-iodide (Figure 2D).

### 3.3. MB-PDT Enhances Markers of Immunogenic Cell Death in Lung Cancer Cells

Calreticulin (CRT) is a marker of DAMPs pathway activation and a signal of “eat me” produced by dying cells and is upregulated under PDT-induced stress. To investigate the effects of MB-PDT on CRT expression we performed immunoblotting. CRT expression was upregulated in MB-PDT-treated H1299 cells 6 and 24 h post treatment compared to untreated cells (Figure 3A). We confirmed CRT upregulation by immunofluorescence in MB-PDT-treated H1299 cells 24 h post-treatment (Figure 3B). Further, to demonstrate specific MB-PDT-induced activation of DAMPs in A549 cells, we analysed expression of the major histocompatibility complex (MHC) class I and intercellular cell-adhesion molecule-1 (ICAM-1) by flow cytometry after 24 and 48 h. MHC-I was upregulated in MB-PDT-treated A549 cells compared to untreated cells after 24 and 48 h. In addition, MB-PDT also induced activation of ICAM-I after 24 and 48 h. (Figure 3C). In Appendix A, the gating strategy was based on MHC-I/(FITC)+ or ICAM-1/(APC)+ in A549 cancer cells. The results indicated that MB-PDT increased the expression of ICD markers in A549 and H1299 cells independently of P53 gene background.

### 3.4. Effects of Non-Autologous Activated CD8^+^ T Cells on Lung Cancer Cells in Killing Assay

Figure 4A summarizes the workflow of the in vitro T cell killing assay. When non-autologous activated CD8^+^ T cells were added to the MB-PDT-treated cancer H1299 cells for 24 h at a ratio of 8:1, a significant decrease in proliferation was detected (Figure 4B). We also investigated the different ratio of effector cells vs. target cells, including a ratio of 2:1 or 6:1 in a co-culture. We observed a significant decreased proliferation of cancer cells in all co-cultures. However, we obtained the best results of killing cancer cells using a ratio of 8:1 in a co-culture. To demonstrate that MB-PDT induced cytotoxic T lymphocyte (CTL) killing in vitro, we incubated activated CD8^+^ T cells with H1299 cells at a ratio of 8:1 for 48 h. Flow cytometry results demonstrated that MB-PDT treatment decreased the viability of lung cancer cells versus untreated cancer cells in a co-culture at 48 h, 4.4% and 44.7%, respectively (Figure 4C, Appendix A). Next, we calculated the MB-PDT specific killing in the same samples using methods previously described [31]. MB-PDT resulted in 66.3% specific killing in co-culture compared to untreated cells. At the same time, the specific killing of MTX, a positive control of ICD, was 80.9% in co-culture compared to untreated cells (Table 1). These results demonstrated that MB-PDT promotes ICD in lung cancer cells in the presence of cytotoxic CD8^+^ T cells.

### 3.5. CD8+ T Cell Reactivity via Granzyme B against Lung Cancer Cells and Organoids

Cancer cell killing by activated CD8^+^ T cells is potentiated by Perforin/Granzyme B activity. To verify the presence of GrzB in our activated CD8^+^ T cells we cultured naive CD8^+^ T cells with hrIL-2 (320 U/mL), anti-CD3 (4 μg/mL) and anti-CD28 (10 μg/mL) antibodies for 24 h at 37 °C, 5% CO_2_ and performed immunofluorescence. We detected high endogenous levels of GrzB in activated CD8^+^ T cells (Figure 5A). Next, we investigated the use of a GrzB fluorescence probe H5 [32] at a final concentration of 5 μM to detect active human endogenous GrzB in a co-culture with CD8+ T cells alongside untreated H1299 cells at a ratio of 6:1, respectively. The time course demonstrated that GrzB probe H5 was activated in a co-culture one hour after adding the probe, and specific fluorescence was still evident 24 h after adding the probe (Figure 5B). The presence of a GrzB inhibitor (10 μM) in a co-culture with CD8+ T cells (vs H1299 cells), at a ratio of 2:1, resulted in a 5.5 fold reduction in fluorescence in untreated cells (Figure 5C). Confocal microscopy demonstrated increased GrzB probe H5 signal in MB-PDT-treated co-cultures compared to co-cultures with untreated H1299 cells alone (Figure 5D). Confocal microscopy revealed a MB-PDT dose dependent signal of GrzB probe H5 one hour post-adding the probe in 3D organoid co-culture. The GrzB signal was specifically detected around the MB-PDT-treated (20 μM or 40 μM) or MTX-treated lung cancer organoids compared to untreated organoids in co-culture (Figure 5E). Organoids derived from two patients were also treated with 20 μM MB-PDT and incubated with activated CD8+ T cells for five days in co-culture, and proliferation of lung cancer organoids was investigated by WST1 assay. 20 μM MB-PDT-treated lung tumour organoids showed a significant decrease of proliferation upon five days of co-culture compared to untreated organoids. MTX (1 μM) also efficiently decreased a proliferation of lung cancer organoids in co-culture (Figure 5F).

## 4. Discussion

MB is a fluorescent dye that has been investigated for many years as an antibacterial and antifungal substance [33,34,35] and antiviral drug in medicine [36,37]. MB generates reactive oxygen species (ROS) in response to light doses lower that 5 J/cm ^2^ and induces ROS dependent apoptosis in cancer cells, including the A549 lung cancer cell line used in this study [30,38]. By combining a clinically viable PS with low energy LEDs, we aimed to determine whether MB-PDT could induce apoptosis and to look for evidence of ICD in lung cancer models. We confirmed that MB-PDT decreases the proliferation of NSCLC cell lines, inhibiting the expression of anti-apoptotic Bcl2 protein and activating caspase 3/7, a marker of apoptosis. To further investigate the therapeutic potential of MB-PDT, we established a more representative 3D human lung cancer organoid model. MB-PDT (20 μM and 40 μM) disrupted organoid structures and induced cell death. Monolayers of cancer cells required lower concentrations of MB and a shorter time for MB-PDT-mediated cell death compered to 3D culture. 

Several studies have demonstrated a strong crosstalk between ICD and long-lasting, protective, anti-tumour immunity. ICD induces endoplasmic reticulum (ER) stress associated with translocation of CRT from the cytoplasm to the cell surface [39,40,41,42,43]. Extracellular CRT serves as an “eat me” signal, stimulating the uptake of dying cells by dendritic cells increasing the productive degradation of tumour cells and presentation of tumour-derived antigens in an immunostimulatory context [43]. Immunoblotting and confocal microscopy demonstrated increased expression of CRT in MB-PDT treated lung cancer cells compared to untreated cells 6 and 24 h post-treatment. CRT-exposure is a canonical indicator of ICD and upregulation of MHC-I and ICAM have also been reported during ICD [44]. Using flow cytometry we detected a 2.7-fold upregulation of ICAM-1 in MB-PDT-treated lung cancer cells compared to untreated cells 24 h post-treatment and 6.4-fold upregulation in MB-PDT-treated cells 48 h post-treatment. It was published that ICAM-1 plays a key role in signal transduction, T-cell stimulation, stabilisation of T-cell receptor-mediated binding between antigen-presenting cells (APC) and T-lymphocytes in ICD-mediated killing [45,46]. Furthermore upregulation of ICAM-1 and MHC-1 expression by A549 cells following sub-lethal irradiation was shown to be associated with enhanced susceptibility to T cell mediated killing [44]. ICD enhances the efficacy of anticancer drugs by promoting antigen uptake, processing and initiation of anti-tumour T cell responses and our data demonstrate that increasing classical markers of ICD in response to MB-PTD treatment could promote this long term efficacy beyond the immediate cytotoxic effects.

Immunogenic cell death following treatment with ionizing radiation promotes anti-tumour T cell responses which can promote the regression of distant lesions not initially exposed to irradiation via a process known as the abscopal effect (reviewed [47]). Cytotoxic CD8^+^ T cells are the targets of immunotherapy, essential to abscopal effects, and are pivotal effector cells for tumour cell clearance [48,49,50,51]. We demonstrated that MB-PDT decreased the proliferation of target monolayer H1299 cells in co-culture with activated CD8^+^ T cells when compared to untreated cells. PDT-associated CTL lysis was enhanced by 66% compared to untreated cancer cells in co-culture with activated CD8^+^ T cells indicating that MB-PDT increased the sensitivity of surviving cancer cells to T cell mediated killing. T cell killing can be mediated by the action of granzyme B, a powerful serine protease released by cytotoxic CD8^+^ T cells [52,53]. To assess the involvement of granzyme B in the elevated level of tumour cell killing seen post MB-PDT, we used a substrate-specific GrzB fluorescent probe H5 [32]. The H5 probe reacts only with active granzyme B (i.e., once it has been transferred from the T cell to the target cell [32]. Both H1299 cells and patient-derived lung cancer organoids demonstrated increased GrzB-mediated killing post MB-PDT Thus, subsequent to its initial cytotoxic effects, cancer cells surviving PDT are rendered more sensitive to T cell mediated killing as has previously been described in cancer cells surviving irradiation [44]. It will be important to investigate the potential of other photosensitizers, such as porphyrins, protoporphyrin, redaporfin, chlorins and phthalocyanines to induce ICD and alter sensitivity to T cell mediated killing. 3D culture models, such as the one used in this study, provide an ideal platform for evaluating these therapeutic approaches.

Several potential mechanisms underlying the MB-PDT-mediated effects on survival of patient-derived cancer organoids remain to be explored. Future work to delineate the correlation between light dose, organoid size, and uptake time of PS, and sensitivity to PDT-induced ICD, are needed to better define the range of therapeutic susceptibility between cancer organoid cultures derived from different donors. Due to the dissociation of organoids following MB-PDT treatment we could discern in this study whether cells at the surface or the organoids were preferentially affected or showed greater signs of ICD than those deeper within the organoid. Given the extent of cell death seen post MB-PDT, it appears that full penetrance was achieved and a fine kinetic examination and dose titration would be needed to investigate this possibility.

Exciting recent advances in the development of PDT are set to increase its range of applications and promote its integration with existing and novel immune therapies. The development of self-exciting “auto-PDT” nanoplatforms for example, where activation is driven by chemiluminescence from compounds co-loaded with the photosensitizer, extends the range of PDT and can begin to overcome barriers associated with low light penetrance into tissues [54]. The application of Near-Infrared thermally activated delayed fluorescence materials with increased efficiency of singlet oxygen generation also extends the range of PDT and is showing promise in preclinical cancer models [55]. Capitalising on the neutrophil influx in response to PDT to target delivery of nano-particles to the tumour illustrates how nanoplatforms can be rationally and effectively combined with knowledge of the biological response to PDT in order to increase its efficiency [56]. In this respect, understanding both how PDT eliminates target cells and how surviving cells are altered in susceptibility to immune-mediated killing can inform the design of therapeutic strategies most effectively combining technology and biology. 

## 5. Conclusions

In the present study, we investigated the effects of MB-PDT on two NSCLC cell lines, H1299 and A549, and primary human lung cancer organoids. Cell proliferation in both cell lines was significantly decreased and associated with activation of caspase 3/7 and a low level of Bcl2 protein after MB-PDT treatment. In H1299 and A549 cells, MB-PDT with low energy light sources rapidly augments ICD markers, such as CRT, MHC-I and ICAM-1 24 and 48 h post-treatment. Our data highlight a marked difference in proliferation between MB-PDT and untreated cancer cells and organoids with increased sensitivity to T cell mediated killing post MB-PD as shown by significantly higher activity of GrzB probe H5 in MB-PDT-treated cells and organoids compared to untreated controls. These data indicate that MB-PDT augments CD8+ T cell-granzyme B mediated cytolysis in both lung cancer cell lines and our lung cancer 3D model. Future studies are required to clarify the mechanism involved in MB-PDT-induced ICD in lung cancer organoids for PDT-based therapy. 

## Figures and Tables

**Figure 1 cancers-14-04119-f001:**
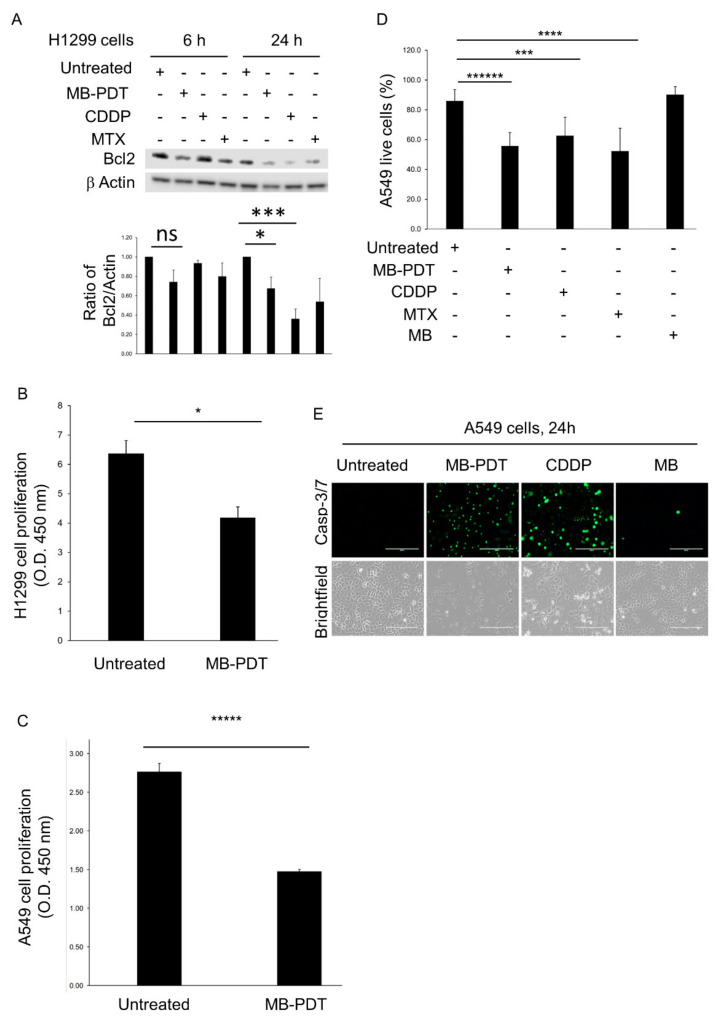
MB-PDT-treated lung cancer cells undergo apoptosis and show reduced proliferation. (**A**) Representative immunoblotting of Bcl2 expression in MB-PDT-treated (20 μM) H1299 cells for 6 and 24 h. Densitometry of three Western blots is presented below. * *p* = 0.03, *** *p* = 0.002. (**B**) The proliferation of H1299 cells was analysed by WST-1 assay 24 h post-treatment. The absorbance was obtained by a Synergy H1 Hybrid Reader at 450 nm. Data included 3 independent experiments, *n* = 9 (per condition), * *p* < 0.05. (**C**) The proliferation of A549 cells was analysed by WST-1 assay 24 h post-treatment. Data included 3 independent experiments, *n* = 8 (per condition). ***** *p* = 0.00001. (**D**) A549 cells were treated with MB-PDT (20 μM). CDDP (20 μM) was used as an inducer of apoptosis, and MXT (1 μM) was used to induce ICD. 20 μM MB without PDT was used as an internal control. Viable cells were determined by trypan blue after 24 h post-treatment. The data are expressed as the mean ± SEM. Data included 2 independent experiments, *n* = 8 (per condition). **** *p* < 0.0078, ****** *p* = 0.000004. (**E**) Representative immunofluorescence images of caspase 3/7 probe activity in A549 cells 24 h post-treatment. 20 μM MB without PDT was used as an internal control. Images taken using an Evos 2 microscope. Scale bar: 200 μm.

**Figure 2 cancers-14-04119-f002:**
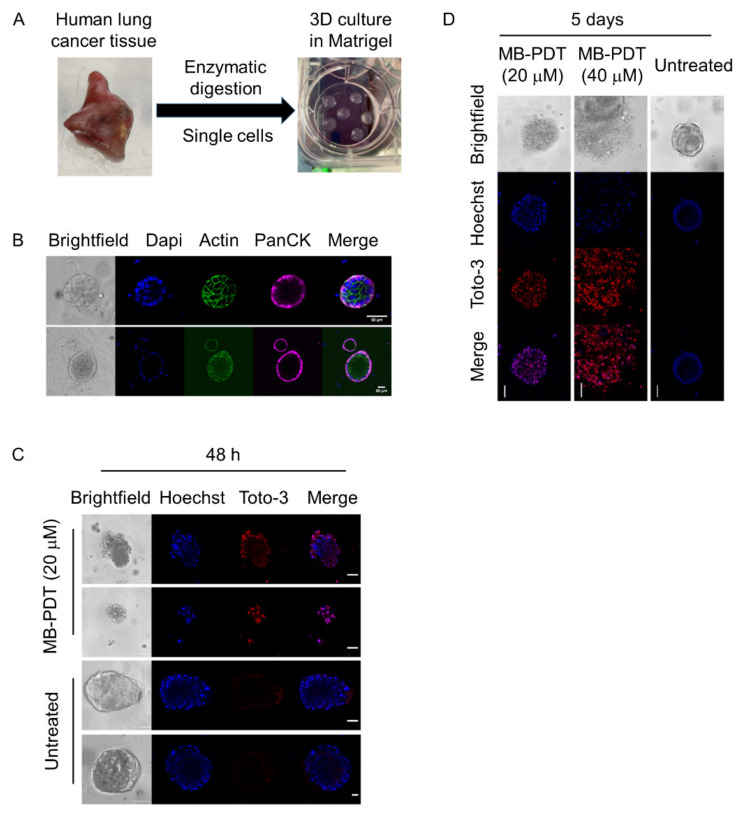
The effects of MB-PDT on apoptosis and morphology in human lung cancer organoids. (**A**) Schematic experimental protocol of human lung cancer 3D culture. (**B**) Representative images of immunofluorescence showing the expression of PanCK, in NSCLC tumour organoids. Images were acquired on a SP8 confocal microscope. Scale bar: 50 μm. (**C**) Representative live images showing different morphology (Brightfield) and dead cells (Toto-3 iodide, red) of 20 μM MB-PDT-treated lung cancer organoids 48 h post-treatment. Scale bar: 45 μm. (**D**) Representative live images of fluorescence of Toto-3 iodide (1 μM) in MB-PDT-treated lung cancer organoids in a dose-dependent manner five days post-treatment. Images were acquired on a SP8 confocal microscope. Scale bar: 45 μm.

**Figure 3 cancers-14-04119-f003:**
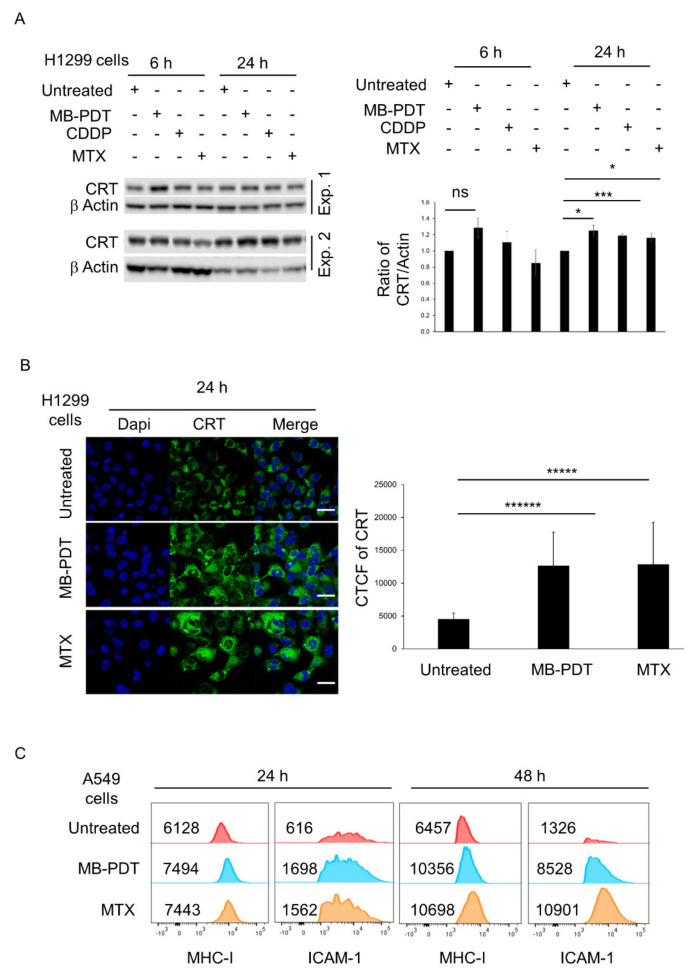
MB-PDT increases the expression of calreticulin and immunogenic cell death markers in lung cancer cells. (**A**) Immunoblotting of CRT expression in H1299 cells treated with MB-PDT (20 μM) for 6 and 24 h. Densitometry of immunoblotting of two experiments is presented on the right. * *p* < 0.05, *** *p* = 0.003. (**B**) Representative immunofluorescence images of CRT in MB-PDT-treated H1299 cells 24 h post-treatment. The corrected total cell fluorescence (CTCF) of CRT is presented on the right. Significance determined by two tailed Student’s *t*-test. ***** *p* = 2.1 × 10^−5^, ****** *p* = 6.9 × 10^−8^. Images were acquired on a SP8 confocal microscope. Scale bar: 45 μm. (**C**) The expression of MHC-I and ICAM-1 was analysed by flow cytometry in A549 cells 24 and 48 h post-treatment. The data are presented in the histograms. The numbers (MFI) indicate the intensity of specific signal.

**Figure 4 cancers-14-04119-f004:**
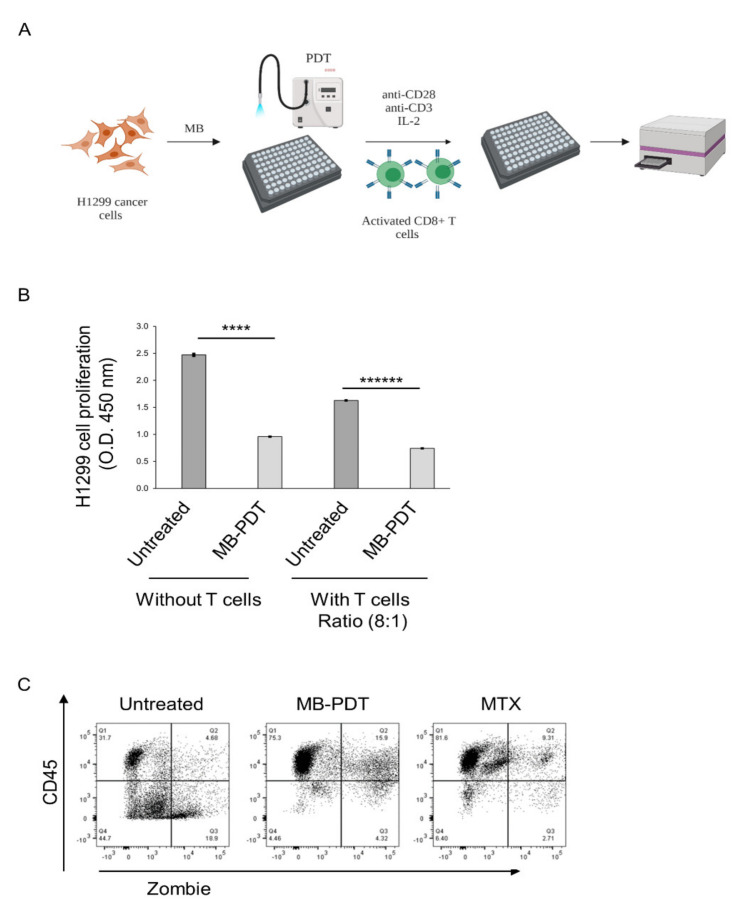
MB-PDT promotes the cytotoxic activity of CD8+ T cells followed by an inhibition of proliferation of human lung cancer cells. (**A**) Schematic experimental protocol of co-culture killing assays. (**B**) H1299 cells proliferation in a co-culture was measured 24 h post-treatment. Then activated CD8^+^ T cells (IL-2 1000 ng/mL) were co-cultured with H1299 cells for 24 h at a ratio of 8:1. The proliferation of H1299 cells in co-culture was analyzed by WST-1 assay. The experiment was carried out in triplicate, and the results presented as means ± SD. Significance determined by two tailed Student’s *t*-test. **** *p* = 0.0002, ****** *p* = 0.000008. (**C**) Flow cytometry confirmed that MB-PDT-treated H1299 cells are killed more efficiently by activated CD8+ T cells in a co-culture for 48 h. A ratio of T cells vs. cancer cells is 8:1. Numbers indicate percentage of specific labelled cells.

**Figure 5 cancers-14-04119-f005:**
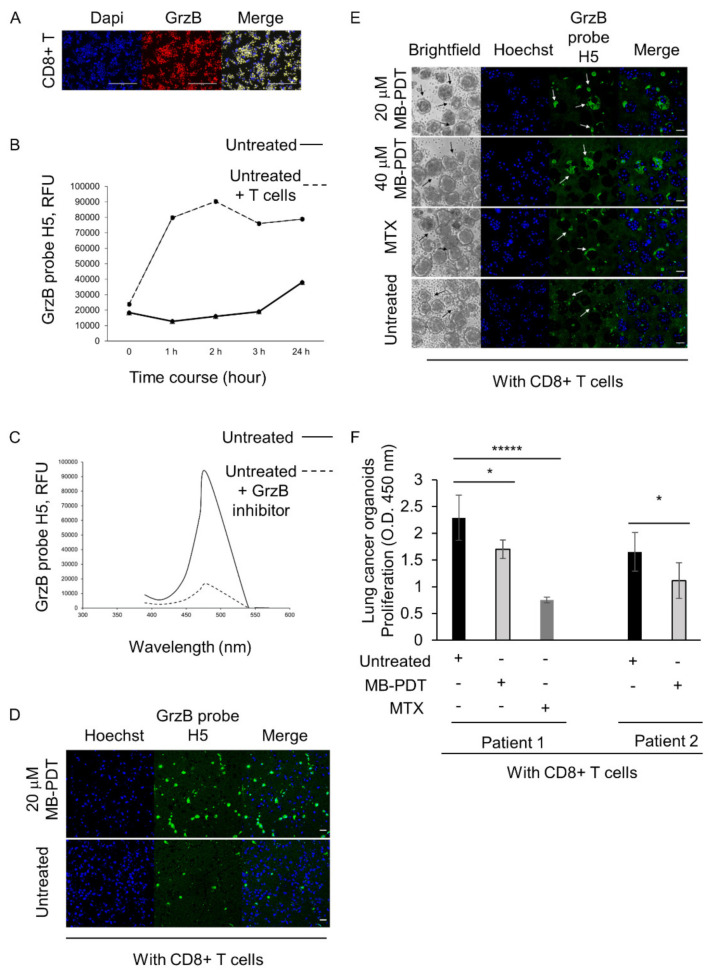
Grz B probe H5 detects the cytotoxic activity of CD8+ T cells in lung cancer cells and organoids co-culture killing assay. (**A**) Representative immunofluorescence images of endogenous Granzyme B in activated CD8+ T cells before adding them into a co-culture with cancer cells. SP8 confocal microscope. Scale bar: 200 μm. Images were processed/analysed with ImageJ. (**B**) The time course of GrzB probe H5 (5 μM) activity followed incubation with activated CD8+ T cells, effectors, and target H1299 cells at a ratio of 6:1. Representative image, *n* = 5 (per condition). The comparison of fluorescence between different samples was obtained by a Synergy H1 Hybrid Reader. The experiment was repeated three times. (**C**) The specific activity of GrzB probe H5 was investigated in a live co-culture with activated CD8+T cells and H1299 cancer cells at a ratio of 2:1, respectively, with or without an inhibitor of GrzB IV (10 μM) added for 24 h. (**D**) Representative live fluorescence images of active GrzB probe H5 (2 μM, in green) in a co-culture with activated CD8+T cells and H1299 cells at a ratio of 2:1, respectively. The imaging was performed 30 min post-adding the probe. Scale bar: 45 μm. (**E**) Representative live fluorescence images of GrzB probe H5 (5 μM) in a co-culture with the activated CD8+ T cells and the human lung cancer organoids at a ratio of 3:1, respectively. The imaging was performed one hour post-adding the probe. Scale bar: 45 μm. (**F**) The proliferation of human lung cancer organoids derived from two different patients was analyzed by WST-1 assay in a co-culture with activated CD8+ T cells. The ratio of T cells vs. cancer organoids 2:1, respectively. MTX (1 μM) was used as a positive control of ICD. The experiment was carried out in five biological samples per condition, and the results presented as means ± SD. * *p* < 0.04, ***** *p* = 0.00004. Significance determined by two tailed Student’s *t*-test.

**Table 1 cancers-14-04119-t001:** Percentage of specific killing in vitro cytotoxicity assay 48 h post-treatment.

Treatment	% of Live Cancer Cells Ratio of T Cells vs. Cancer Cells (0:1)	% of Live Cancer Cells Ratio of T Cells vs. Cancer Cells (8:1)	The Percentage of Specific Killing
Untreated *	80.2	44.7	n/a
MB-PDT	23.7	4.4	66.3
MTX	60.3	6.4	80.9

* Activated CD8+ T cells were co-cultured with H1299 cells with/without treatment for 48 h. Then, the relative frequency of the remaining target cells was quantified by flow cytometry. Pre-gating was done on APC cy7 CD45− (cancer) cells and APC cy7 CD45+ (T cells). A gating strategy is illustrated in Appendix A. Data of flow cytometry were acquired on 5 L LSR flow cytometry. To calculate the percentage of specific killing, the following formula [23] was used: % specific kill = 100–[100 × (live MB-PDT-treaded cancer cells in co-culture/live untreated cancer cells in co-culture)/(live MB-PDT-treated cancer cells/live untreated cancer cells)].

## Data Availability

The data presented in this study are available on request from the corresponding author.

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
