# Peer review of "Phototherapeutic Induction of Immunogenic Cell Death and CD8+ T Cell-Granzyme B Mediated Cytolysis in Human Lung Cancer Cells and Organoids"

_cancers, 2022, doi:10.3390/cancers14174119_

Round 1

Reviewer 1 Report

This research is the investigation of PDT efficacy by using the 3D culture. This study is interesting, but the description or sentences are too poor to understand the study completely.
Taken together, major revisions should be made before re-submission. The paper would be re-considered only when all the comments were responded.

1. Introduction

There is no paragraph to introduce or examine the 3D cell culture, although the concept is essential for this study. The authors should add the sentences for the concept and recent research by using 3D culture to evaluate cancer characteristics for therapy. After the sentences, the authors can introduce your research. At the current version, the reviewer cannot recommend the publication. To reduce the authors’ burden, I suggest at least these references to be added for the revision.

Review (for concept)

doi.org/10.1016/j.ddtec.2017.03.002

Cancers 202012(10), 2754

Research

Proliferation

doi.org/10.1016/j.biomaterials.2018.10.014

Alternation

Cells 202211(2), 305

Invasion
Tissue Eng. Part C Methods
 201925, 711–720
 https://doi.org/10.1089/ten.tec.2019.0189

Morphology

doi.org/10.1073/pnas.1719405115

Drug resistance

doi.org/10.1016/j.biomaterials.2010.07.064

2. lines 71

The authors should mention the difference between the two cell lines.

3. Line 104

Did the authors check the appropriate light dose? Figure 2D shows the results, but the information is poor.

4. Discussion

This part can not be understood. The authors should discuss the strength by quoting related papers above. At the current version, the reviewer cannot recommend the publication.

5.

The effect of MB-PDT on the surface of organoids must be different from the inner. The authors should discuss the points.

Author Response

Dear Reviewer 1,

Thank you for your assistance and guidance through the review process. We greatly appreciate the reviewers’ constructive criticism. We are pleased to resubmit the revised version of our manuscript “Phototherapeutic induction of immunogenic cell death and CD8+ T cells-granzyme B mediated cytolysis in human lung cancer cells and organoids” (#1782902). Please find attached a revised copy of our manuscript (#1782902) modified in response to the reviewers comments.

We thank the reviewers for their time and their insightful suggestions, which have helped us improve the manuscript. We have addressed all points raised by the reviewer 1 and a point by point synopsis of our response follows below:

Reviewer 1:

Comment 1:

There is no paragraph to introduce or examine the 3D cell culture, although the concept is essential for this study. The authors should add the sentences for the concept and recent research by using 3D culture to evaluate cancer characteristics for therapy. After the sentences, the authors can introduce your research.

Response: We have added a paragraph which introduces the concept of 3D versus 2D culture models in screening for anti-cancer drugs (lines 47-64). Therein we have included the recommended references, where appropriate, and hope this now provides a suitable introduction to 3D culture, before outlining our experimental work as suggested.

Comment 2:

The authors should mention the difference between the two cell lines.

Response: We have added to the introduction of the two cell lines additional detail beyond their difference in p53 status and added a reference highlighting their differential sensitivity to DNA damage in lines 91-94.

Comment 3:

Did the authors check the appropriate light dose?

Response: We had determined in prior experiments that a light dose of 10 j / cm2 delivered via LED did not affect the viability of the cell lines used. We have added a line to indicate this at line 131-132 and cite in support of this published evidence from the results of Lim et al who demonstrated that doses of 30-60 J / cm2 did not impact the viability of A549 cells. The capacity of the light dose used to activate the photosensitizer (resulting in decreased viability and increased caspase activity) is shown in Fig. 1D and 1E by the comparison between MB only and MB-PDT. As this light dose proved effective at inducing cell death and markers of ICD it was kept constant in other experiments.

Comment 4:

Relates to the discussion: This part can not be understood. The authors should discuss the strength by quoting related papers above. 

Response: We have made extensive changes to the discussion in response to this comment and restructured it to clarify it’s themes. We have built on the stronger introduction to 3D culture systems now provided in the introduction and proceed to review our results, we then extended the description referring to the measures and significance of immunogenic cell death in cancer treatments. We also expanded the section on the importance of altered sensitivity to T cell mediated killing. We then close the discussion with allusion to recent developments PDT and the need for increased mechanistic understanding to inform therapeutic design. Please see the various additions between lines 448 and 526.

Comment 5:

The effect of MB-PDT on the surface of organoids must be different from the inner. The authors should discuss the points.

Response: This would indeed be of interest however we were not able to determine differences in responsiveness to MB-PDT as relates to the original position of the cells in the organoids as dissociation of the organoids occurred post treatment. We have added extra text to clarify this and identify it as an area worthy of future study in new lines 507-512.

“Due to the dissociation of organoids following MB-PDT treatment we could discern in this study whether cells at the surface or the organoids were preferentially affected or showed greater signs of ICD than those deeper within the organoid. Given the extent of cell death seen post MB-PDT it appear full penetrance was achieved and a fine kinetic examination and dose titration would be needed investigate this possibility.”

Reviewer 2 Report

Photodynamic therapy and the corresponding immunogenic cell death (ICD) are playing an important role in cancer synergistic treatment. The authors reported the augmented antitumor effect of ICD in response to methylene blue photodynamic therapy (MB-PDT) in human non-small lung cancer. Overall, the experiments were done substantially and systematically, the discussions herein are reasonable. Therefore, this manuscript could be published after minor revisions noted below.

1. The authors should provide results on ROS generation capability at cancer cell level to prove the photodynamic performance of methylene blue.

2. When talking about the significance of ICD or PDT for the treatment of cancer, some recent representative references should be cited: Exploration 2022; 20210166, DOI: 10.1002/EXP.20210166; Small 2022, 18, 2106215. DOI: 10.1002/smll.202106215; Adv. Sci. 2021, 8, 2102970. DOI: 10.1002/advs.202102970.

3. The authors used the LED as an excitation source for MB-PDT, even though they claimed it is a low-energy LED, the cytotoxicity of the excitation light itself should also be evaluated on different cell lines.

Author Response

Dear Reviewer 2,

Thank you for your assistance and guidance through the review process. We greatly appreciate the reviewers’ constructive criticism. We are pleased to resubmit the revised version of our manuscript “Phototherapeutic induction of immunogenic cell death and CD8+ T cells-granzyme B mediated cytolysis in human lung cancer cells and organoids” (#1782902). Please find attached a revised copy of our manuscript (#1782902) modified in response to the reviewers comments.

We thank the reviewers for their time and their insightful suggestions, which have helped us improve the manuscript. We have addressed all points raised by the reviewer 2 and a point by point synopsis of our response follows below:

Reviewer 2

Comment 1: The authors should provide results on ROS generation capability at cancer cell level to prove the photodynamic performance of methylene blue

Response: We demonstrated the photodynamic performance of methylene blue in the requirement for light to stimulate it’s cytotoxic activity, as evidenced by increased cell death and apoptosis, following exposure to MB+Light exposure versus MB alone in the data shown in Fig. 1D and 1E. As the reviewer indicates ROS production has an important role in driving apoptosis post PDT and we have highlighted this with a citation showing treatment with lower levels of light than used in our study induce ROS generation by MB treated A549 cells in lines 448-450.

“MB generates reactive oxygen species (ROS) in response to light doses lower that 5 J / cm 2 and induces ROS dependent apoptosis in cancer cells, including the A549 lung cancer cell line used in this study [38],[30].”

Comment 2:

When talking about the significance of ICD or PDT for the treatment of cancer, some recent representative references should be cited: Exploration 2022; 20210166, DOI: 10.1002/EXP.20210166; Small 2022, 18, 2106215. DOI: 10.1002/smll.202106215; Adv. Sci. 2021, 8, 2102970. DOI: 10.1002/advs.202102970.

Response: We have added a paragraph to the discussion referring to recent developments in PDT related to cancer treatment and cited the suggested references (where it was appropriate to do so). Please see discussion lines 513 to 526.

Comment 3:

The authors used the LED as an excitation source for MB-PDT, even though they claimed it is a low-energy LED, the cytotoxicity of the excitation light itself should also be evaluated on different cell lines

We determined in prior experiments that a light dose of 10 J / cm2 delivered via LED did not affect the viability of the cell lines used. To clarify this in the text we have added a line at 131-132 and cite the published evidence that light treatments at doses of 30-60 J / cm2 (i.e. 3-6 times greater than we used in the current study) does not impact the viability of A549 cells (Lim et al. Ref 30).

Again, we thank the reviewers and the corresponding editor for the opportunity to refine our manuscript in response to the reviewers’ comments and hope they find our revised manuscript improved.

Sincerely yours.

Round 2

Reviewer 1 Report

It is crucial to investigate the treatment efficiency of inner tissues because tumor tissue is complicated. If the evaluation is not performed, research using 3D spheroids is not very important. 

If the authors can not, the manuscript is not worth publishing.

Author Response

We respectfully highlight the fact that our study describes experiments using organoids containing various cell types reflecting the complexity of the tumour tissue and not simple spheroids. We believe patient-derived cancer organoids, as a multicellular 3D structures, recapitulate the cancer tissue architecture and maintain the functions of tumour tissue during-long-term expansion in vitro and serve as robust models for research studies and therapeutic approaches. Patient-derived cancer organoids provide a potential alternative to animal experiments.